# Differential Dependency of Human Pancreatic Cancer Cells on Targeting PTEN via PLK 1 Expression

**DOI:** 10.3390/cancers12020277

**Published:** 2020-01-23

**Authors:** Jungwhoi Lee, Jungsul Lee, Woogwang Sim, Jae-Hoon Kim

**Affiliations:** 1Department of Applied Life Science, SARI, Jeju National University, Jeju-do 63243, Korea; 2Department of Bio and Brain Engineering, KAIST, Daejeon 34141, Korea; jungsullee@gmail.com (J.L.); zoshsxla@gmail.com (W.S.); 3Subtropical/tropical Organism Gene Bank, Jeju National University, Jeju-do 63243, Korea

**Keywords:** PTEN, PLK1, pancreatic cancer, prognosis, companion biomarker

## Abstract

Even though the tumour suppressive role of PTEN is well-known, its prognostic implications are ambiguous. The objective of this study was to further explore the function of PTEN expression in human pancreatic cancer. The expression of PTEN has been dominant in various human cancers including pancreatic cancer when compared with their matched normal tissues. The pancreatic cancer cells have been divided into PTEN blockade-susceptible and PTEN blockade-impassible groups dependent on targeting PTEN by altering intracellular signaling. The expression of *PTEN* has led to varying clinical outcomes of pancreatic cancer based on GEO Series (GSE) data analysis and Liptak’s z analysis. Differential dependency to PTEN blockade has been ascertained based on the expression of polo-like kinase1 PLK1 in pancreatic cancer cells. The prognostic value of PTEN also depends on PLK1 expression in pancreatic cancer. Collectively, the present study provides a rationale for targeting PTEN as a promising therapeutic strategy dependent on PLK1 expressions using a companion biomarker discovery platform.

## 1. Introduction

Pancreatic ductal-adenocarcinoma, which is the most common but deadly form of human pancreatic cancers, is highly dangerous and associated with poor survival rates [1]. Although continuous efforts have been made during the last several decades to improve its prognosis, the incidence rate of pancreatic cancer is almost equal to its death rate [2]. Such prognosis of pancreatic cancer prompts us to exploit an innovative therapeutic strategy for a pancreatic cancer blockade. 

Several studies have reported that phosphatase and tensin homologue (*PTEN*) mutated on chromosome 10 has tumour suppressive functions in nearly all malignant cancers [3,4]. However, PTEN has been increasingly involved in non-tumour suppressive roles, which provides a strong rationale underlying its biological significance in cancer [5,6,7,8,9].

Polo-like kinase 1 (PLK1) is a critical regulator of the cell cycle and carcinogenesis [10,11]. In addition, it has been suggested as a diagnostic biomarker and an anti-cancer target [10,12]. Therefore, PLK1 inhibitors are under clinical investigation in various cancers [13]. Nonetheless, the mechanism underlying PLK1 inhibition in pancreatic cancer remains unclear.

Currently, most cancer patients receive anti-cancer therapies such as chemotherapy, hormone therapy, biological therapy, or combinations thereof [14]. Although these anti-tumour treatments are utilized, they have been increasingly needed by cooperating with appropriated tools, such as the companion biomarker. Even though companion biomarkers might be used to lead treatment for different diseases, their role in malignant tumours is limited by their ineffectiveness, basal toxicity, and the high cost of cancer treatment [15,16]. Therefore, effective collaboration of anti-cancer therapy and companion-biomarkers is urgently required.

Considering the ambiguous function of PTEN in the tumour microenvironment, the biological significance of PTEN expression requires re-evaluation for cancer therapies integrated with companion biomarkers. In the present study, we demonstrate the effect of PTEN blockade on human pancreatic cancer based on PLK1 expression in vitro and in vivo.

## 2. Results

### 2.1. Expression of Phosphatase and Tensin Homologue (PTEN) in Human Pancreatic Cancer

We have used the *Oncopression* database (https://www.oncopression.com) to determine the mRNA expression profiles of *PTEN* in human cancers and their matched normal tissues. Cancers of the brain, stomach, head and neck, lung, ovary, pancreas, prostate, and kidney express significantly higher levels of *PTEN* mRNA when compared to their matched normal tissues. In contrast, colon, skin, and thyroid cancers express significantly less *PTEN* mRNA than their matched normal tissues (Figure 1a). To verify the differential expression of *PTEN* in various human cancers, we investigated the protein expression of PTEN in various normal and cancer tissues. Cancers of brain, colon, kidney, pancreas, spleen, bladder, breast, cervix, prostate, and testis expressed higher PTEN protein than their matched normal tissues but not in liver and uterine cancers. Cancers of the brain, colon, and pancreas have shown remarkable expression, and pancreatic cancer displayed the most abundant PTEN expression levels compared with other cancer types (Appendix A and Figure 1b). To ascertain the expression of PTEN and its involving signaling molecules including p-PTEN and AKT in human pancreatic cancer, we have performed Western blot analysis using separate human pancreatic cancer and normal tissues. Phoshor-PTEN, PTEN, and Phoshor-AKT have been highly expressed in pancreatic cancer tissues when compared with a normal pancreas (Figure 1c). 

### 2.2. In Vitro Effects of PTEN Inhibition 

To determine the functional role of PTEN expression in human pancreatic cancers, we first evaluated the endogenous expression of PTEN in six pancreatic cancer cell lines and normal pancreatic duct epithelial H6c7 cells by Western blot analysis. PTEN was abundantly expressed in both human pancreatic cancer cells and H6C7 cells (Figure 2a). Next, we determined the functional role of PTEN expression by targeting PTEN signaling using SF1670, which is a pharmacological PTEN inhibitor. Exposure to 0.5 µM of SF1670 has resulted in approximately 1.19-fold, 1.15-fold, and 1.10-fold increased viability of AsPC-1, Capan-2, and SNU-213 cells, respectively, when compared with the control (Figure 2b). In contrast, the same treatment decreased the viability of CFPAC-1, Panc-1, and Miapaca-2 cells by approximately 22%, 29%, and 29%, respectively. To determine the effect of targeting PTEN on viability, we have investigated the levels of proliferating cell nuclear antigen (PCNA) in AsPC-1 cells, which were most increased on viability by SF1670 treatment. We also determined the caspase-3 involved apoptosis of Panc-1 cells. The abundance of PCNA was increased by SF1670 treatment dose-dependently in AsPC-1 cells (Figure 2c). SF1670 treatment induced the cleavage of caspase-3, which was effectively suppressed upon pre-treatment with a pan-caspase inhibitor known as z-VAD (Figure 2d). In addition, SF1670 treatment has altered cell migration similarly in human pancreatic cancer cells (Figure 2e). To further elucidate these effects of targeting PTEN signaling, we ablated PTEN expression via si-PTEN transfection in six human pancreatic cancer cell lines and H6c7 cells. Silencing PTEN expression significantly increased the viability and migration of H6c7, AsPC-1, Capan-2, and SNU-213 cells, respectively. In contrast, the same treatment significantly decreased the viability and migration of CFPAC-1, Panc-1, and Miapaca-2 cells, respectively (Appendix A, Figure 2f,g). Similar results have been obtained using different si-PTEN oligonucleotides (Appendix A). We have also verified the effects of PTEN blockade using SF1670 treatment in H6c7 cells. Upon SF1670 treatment, the viability and migration of H6c7 cells were similar to that of AsPC-1, Capan-2, and SNU-213 cells, respectively (Appendix A). According to the response by targeting PTEN, AsPC-1, Capan-2, and SNU-213 cells were classified as a PTEN blockade-impassible group and CFPAC-1, Panc-1, and Miapaca-2 cells were divided into the PTEN blockade-susceptible group.

### 2.3. Effect of PTEN Inhibition on the Intracellular Signaling Pathway 

To delineate the mechanism of the different responses to the PTEN blockade in human pancreatic cancers, we have analyzed the signal transduction pathways using a phospho-RTK array. PTEN knockdown reduced the levels of hepatocyte growth factor receptor (HGFR) phosphorylation in Panc-1 cells when compared with control si-RNA transfected cells (Figure 3a). To further analyze the detailed mechanism of PTEN ablation, the expression levels of HGFR-related intracellular molecules [17,18] were evaluated following si-PTEN transfection in H6c7, AsPC-1, and Panc-1 cells. PTEN silencing increased the expression of p-HGFR (Y1234/1235), HGFR, phosphor-AKT (S473), and phosphor-FAK (Y397) in H6c7 and PTEN-blockade impassible (P-I) AsPC-1 cells compared with control si-RNA transfection. In contrast, ablation of PTEN expression has decreased the expression of p-HGFR (Y1234/1235), HGFR, phosphor-AKT (S473), and phosphor-FAK (Y397) in PTEN-blockade susceptible (P-S) Panc-1 cells when compared with control si-RNA transfection (Figure 3b).

### 2.4. Correlation Between Polo Like Kinase1 (PLK1) Expressions and Responses to PTEN Regulation in Human Pancreatic Cancer

To determine the prognosis of patients with pancreatic cancer based on PTEN expression, we have analyzed GEO data-sets such as PAAD-US-TCGA, GSE78229, GSE62452, PACA-AU, and GSE21501 in individuals. Prognosis to PTEN expression in pancreatic cancer patients has been strikingly distinct. The elevated PTEN expression had an adverse effect on pancreatic cancer prognosis in PAAD-US-TCGA, GSE78229, and GSE62452 data-sets (Figure 4a). In contrast, increased PTEN expression favorably altered the pancreatic cancer prognosis in PACA-AU (TNM:T = 3 and Grade = 2) and GSE21501 data sets (Figure 4b). To validate the factors underlying the differential response to the PTEN blockade and the conflicting prognosis to PTEN expression, the iterative patient partitioning (IPP) method has been used. PLK1 has been selected as the most potent gene based on the IPP scores (Table 1). Western blot analysis was used to correlate the endogenous expression of PLK1 with responses to PTEN regulations including the following candidates such as the GATA binding protein 3 (GATA3), cyclin-dependent kinase14 (CDK14), and LON peptidase N-terminal domain and ring finger 2 (LONRF2). The expression of PLK1 has been uniquely associated with the differential response of human pancreatic cancer cells to PTEN regulation. Expression of PLK-1 was relatively high (1< PLK1/GAPDH) in PTEN-blockade susceptible CFPAC-1, Miapaca-2, and Panc-1 cells when compared with the levels (1> PLK1/GAPDH) in PTEN-blockade impassible AsPC-1, Capan-2, SNU-213, and H6c7 cells (Figure 4c). To ascertain the correlation between PLK1 expression and responses to PTEN regulation, we investigated PLK1 over-expression or knockdown. Transient transfection of overexpressing PLK1 induced the opposite response to treatment with dose-dependent SF1670 in PTEN-blockade impassible AsPC-1 and SNU-213 cells compared with that of control vector-transfected cells (Figure 4d). In addition, cells transfected with PLK1-specific siRNA showed resistance to dose-dependent SF1670 treatment in PTEN-blockade susceptible Miapaca-2 and Panc-1 cells compared with the scrambled siRNA-transfected cells (Figure 4e). We further investigated the signal transduction events that were specifically activated by PLK1 overexpression or knockdown. Enhanced PLK1 expression decreased the levels of AKT phosphorylation and survivin expressions in PTEN-blockade impassible (P-I) AsPC-1 and SNU-213 cells independent of PTEN expressions (Figure 4f and Appendix A). In contrast, transfection of si-PLK1 induced the expression of phosphor-AKT and survivin in PTEN-blockade susceptible (P-S) Miapaca-2 and Panc-1 cells compared with those of scrambled siRNA-transfected cells independent of PTEN expressions (Figure 4g and Appendix A).

### 2.5. In Vivo Effects of PTEN Inhibition 

To validate the effects of PTEN inhibition in vivo, we have developed PTEN-blockade susceptible Panc-1 and PTEN-blockade impassible AsPC-1 xenograft models. Tumour-laden mice were subcutaneously injected with the control buffer, 10 mg/kg, or 30 mg/kg SF1670 when tumours reached an average size of approximately 150 mm^3^. Panc-1 xenograft tumours treated with the control buffer grew to an average size of 382.53 ± 100.472 mm^3^ in 40 days after transplantation while those treated with 10 mg/kg and 30 mg/kg SF1670 grew to an average size of 230.29 ± 13.25 mm^3^ and 121.76 ± 46.29 mm^3^, respectively, in 40 days post-transplantation. There has been no significant difference in weight loss (Figure 5a). In contrast, treatment with the control buffer, 10 mg/kg, or 30 mg/kg SF1670 had no effect in AsPC-1 xenograft models. No weight loss has been detected in AsPC-1 xenograft models (Figure 5b). To ascertain the anti-growth effect of PTEN inhibition, phosphor-AKT, phosphor-FAK (Y397), and caspase-3-mediated apoptosis in SF1670-treated Panc-1 xenograft models has been investigated. Treatment with 10 mg/kg and 30 mg/kg of SF1670 inhibited phosphor-AKT and phosphor-FAK (Y397) in PTEN-blockade susceptible Panc-1 xenograft models (Figure 5c). In addition, treatment with 10 mg/kg and 30 mg/kg of SF1670 induced gradual cleavage of caspase-3 in PTEN-blockade susceptible Panc-1 xenograft models (Figure 5d).

### 2.6. Prognosis Associated with PLK1 Expression in Human Pancreatic Cancer 

To further validate the role of PLK1 expression in the prognosis of patients with pancreatic cancer, the prognostic value of the GSE17891 data-set without any statistical significance to a rise in PTEN expression has been analyzed using Kaplan-Meier curves. Elevated PTEN expression has been specifically associated with a poor prognosis of pancreatic cancer in PLK1-expressing GSE17891 data set compared with a PLK1 non-expressing data set (Figure 6a,b). Additionally, *PLK1* mRNA levels in high PTEN-expressing group have been significantly up-regulated compared with those in low PTEN-expressing group in GSE78229 and GSE62452 data sets, which had an adverse prognosis of pancreatic cancer patients to elevated PTEN expression (Figure 6c). In contrast, *PLK1* mRNA levels in a high PTEN-expressing group have been significantly down-regulated when compared with the low PTEN group in PTEN-expressing PACA-AU (TNM:T = 3, Grade = 2) and GSE21501 data sets, which had a favorable prognosis of pancreatic cancer patients to elevated PTEN expression (Figure 6d). In addition, we investigated whether soluble PLK1 may be detected in whole blood samples of pancreatic cancer patients to stratify patients for PTEN targeting. Soluble PLK1s have been detected in whole blood samples obtained from pancreatic cancer patients (Figure 7a, 6/12 strong positive, 3/12 moderate positive, and 3/3 similar to control) compared with healthy donors using a sandwich-ELISA (Appendix A). Soluble PLK1 has also been clearly detected in whole blood samples of pancreatic cancer patients when compared with healthy donors using Western blot analysis (Figure 7b).

## 3. Discussion

Understanding the etiology of pancreatic cancer has become increasingly important due to its peculiar clinical prognosis. Evidently, the overall survival rate of patients diagnosed with pancreatic cancer has never improved in the last few decades, with almost equal rates of incidence and death [19,20]. 

PTEN has been explored as a tumour suppressor in nearly all malignant cancers [3,4,21]. Despite increasing the features of PTEN as a non-tumour suppressor in various cancers, PTEN has still been the focus of intervention in malignancy [5,6,7,8,9]. Presently, we revealed a chimerical expression pattern of PTEN in various normal and cancer tissues. Various cancers, including pancreatic cancer, have shown higher expression levels of PTEN protein compared with their matched normal tissues excluding lung and uterine cancers, even though mRNA levels of *PTEN* varied in normal and cancer tissues. The expression pattern of PTEN in pancreatic cancer also showed varying patterns in vitro and in vivo, which indicated that the function of PTEN should be separated from its classical role as a tumour suppressor after disrupting homeostasis. 

In agreement with previous studies about passing the tumour suppressive function of PTEN [5,6,7,8,9], pancreatic cancer cells have been grouped into PTEN-blockade impassible and PTEN-blockade susceptible categories based on our in vitro results. Intuitively, this result suggests that targeting PTEN might be a potential anti-cancer strategy in case of a PTEN-blockade susceptible group. Of interest, a blockade of PTEN has shown a critical rationale beyond other anti-pancreatic cancer strategies. Treatment with chemotherapeutics such as gemcitabine, FOLFIRINOX, and nanoparticle albumin-bound paclitaxel (nAb-PTX) have been reported to induce severe side effects including anemia, depilation, diarrhea, and vomiting in almost all patients [22,23,24,25,26,27,28,29]. In our study, we demonstrated that targeting PTEN had no basal cytotoxicity on the human pancreatic duct epithelial cell line H6c7, which induced the viability and migration features of H6c7 cells. This implied that the PTEN blockade has potentially no basal toxicity for normal pancreatic cells.

The receptor tyrosine kinase HGFR is frequently overexpressed in pancreatic cancer and has been reported in drug resistance, metastasis, and angiogenesis in human pancreatic cancer [30,31]. Therefore, HGFR signaling is considered a prominent therapeutic target in pancreatic cancer [27]. Given that HGFR expression is regulated by silencing PTEN expression, our results are consistent with a previous study [32]. However, the present study particularly showed that ablation of PTEN expression has decreased HGFR expression and its related signaling such as phosphor-AKT and phosphor-FAK levels in the PTEN-blockade susceptible group. These results indicate that the differential response of human pancreatic cancer cells to PTEN blockade is consistent with the intracellular mechanism underlying the altered HGFR expression. In addition, none of the effective inhibitors or antibodies against HGFR signaling have shown any benefit in clinical trials to date [27]. Based on altered HGFR signaling by targeting PTEN, the blockade of PTEN should be seriously considered as a strategy for the inhibition of HGFR signaling in a PTEN-blockade susceptible group. Notably, different alteration of HGFR, p-AKT, and p-FAK expression by targeting PTEN in both a PTEN blockade impassible group or a susceptible group should be further investigated to establish the fundamental signaling pathway in pancreatic cancer cells.

The PTEN blockade induced a different response in human pancreatic cancer in vitro, even though various pancreatic cancer cells have shown substantially similar PTEN expression. In addition, the clinical outcomes of pancreatic cancer patients also differed strikingly independent of PTEN expression patterns, which implies that there might be the role of specific factors. To evaluate this hypothesis, we conducted an IPP method and nominated PLK1 as the critical regulator of PTEN-blockade impassible or susceptible groups. According to the alterations of the viability feature and intracellular signaling to the PTEN blockade following the over-expression or ablation of endogenous PLK1, we have suggested a novel function of PLK1 in human pancreatic cancer. However, further study should be performed to validate the correlation between PTEN and PLK-1 expressions in human pancreatic cancer cells.

In vivo anti-cancer effects of PTEN targeting have been observed in PTEN-blockade susceptible Panc-1 xenograft models in each group (control: *n* = 6, SF1670: *n* = 12). Importantly, administration of SF1670 showed no induction of the tumour size in PTEN-blockade impassible AsPC-1 xenograft models unlike in vitro results, which suggests the safety of PTEN targeting as a therapeutic strategy. No unexpected side effects have been found in both a PTEN-blockade impassible group and the surrounding normal cells. The findings may facilitate patient-specific therapeutic strategies targeting PTEN signaling. 

Clinical evidence has also strongly indicated the critical role of PLK1 expression in pancreatic cancer prognosis based on PTEN expression. In the GSE17891 data-set, the prognostic value of PTEN expression was insignificant, even though there was deterioration in overall survival in the highly expressing PTEN group compared with the group with a low expression of PTEN regardless of PLK1 expression. Elevated expression of PTEN and expression of PLK1 have significantly aggravated the prognosis of pancreatic cancer patients, which suggests that PLK1 is a critical “signal informer” defining the clinical significance of PTEN expression. In addition, the dual expression of high levels of PLK1 and PTEN has aggravated the clinical outcomes of pancreatic cancer patients in GSE78229 and GSE62452. These results indicate that PLK1 is a bi-functional molecule acting as a critical regulator and biomarker of advanced malignancy of pancreatic cancer involved in PTEN expression. However, rephrasing differently, the high expression of PLK1 may represent an Achilles’ tendon for treating pancreatic cancer targeting the PTEN strategy. To the best of our knowledge, this is the first report showing that the expression of PTEN and PLK1 are companion biomarkers associated with an alternative therapeutic strategy and prognostic significance in human pancreatic cancer. Advances in the development of companion biomarkers in the tumour microenvironment have been increasingly reported in recent years [15,33]. In the treatment of various malignant tumours, the value of companion biomarkers has already been established, which suggests their role as an innovative tool in drug development and personalized treatment in clinical trials. Nonetheless, the applications of companion biomarkers in tumour oncology are still limited [34]. In the long term, single-drug tests may be replaced by multi-drug tests guided by proper companion biomarkers. Controlled trials of these techniques are needed to develop rational and cost-effective therapeutic strategies addressing the needs of individual patients and the cancer treatment system [16,35,36]. Currently, the expression of PTEN and PLK1 are clearly based on in vitro and in vivo clinical evidence. Based on the innovative correlation between PTEN and PLK1, detecting soluble PLK1 using the whole blood samples from pancreatic cancer patients and accessible techniques such as Western blots and ELISA might be an innovative therapeutic strategy with a blockade of PTEN for patients already diagnosed with pancreatic cancer or such patients who have exhausted all other options. Follow-up studies using a wide array of whole blood samples are absolutely needed to establish PLK1 as a companion biomarker with PTEN.

Collectively, this study provides a rationale for targeting PTEN as an innovative therapeutic strategy based on the expression of PLK1 in pancreatic cancer patients.

## 4. Materials and Methods

### 4.1. Gene Expression Analysis

Microarray expression profiles have been obtained from the *Oncopression* database (www.oncopression.com) as previously described [37].

### 4.2. Cell Culture and Reagents

AsPC-1(#21682), Capan-2(#30080), Miapaca-2(#21420), Panc-1(#21469), and SNU-213(#00213) cells were purchased from the Korean Cell Line Bank (KCLB, Seoul, Korea). CFPAC-1(CRL-1918) was obtained from the American Type Culture Collection (ATCC, Manassas, VA, USA). The cells were grown as previously described [38]. H6c7(ECA001) has been obtained from Kerafast (Boston, MA, USA) and grown as described previously [39]. Antibodies to PTEN (#9188), phosphor-PTEN (#9551), hepatocyte growth factor receptor (HGFR, #8198), phosphor-HGFR (Tyr1234/1235, #3077), focal adhesion kinase (FAK, #3285), phosphor-FAK (Tyr397, #3283), protein kinase B (AKT, #4691), phosphor-AKT (Ser473, #4060), proto-oncogene tyrosine-protein kinase Src (Src, #2108), phosphor-Src (Tyr416, #6943), polo-like kinase 1 (PLK1, #4513), proliferating cell nuclear antigen (PCNA, #2586), survivin (#2808), and glyceraldehyde-3-phosphate dehydrogenase (GAPDH, #5174) have been purchased from Cell Signaling Technology (Beverly, MA, USA). SF1670 has been purchased from Sigma-Aldrich (St. Louis, MO, USA). Whole blood samples of healthy donors and pancreatic cancer patients have been ordered from innovative research (Novi, MI, USA). Separate human pancreatic cancer and normal pancreatic tissue lysates were purchased from abcam (Cambridge, UK, normal pancreas: ab29816, pancreatic cancer tissue: ab29817) and Prosci Inc. (Poway, CA, USA, normal pancreas: #1307, pancreatic cancer tissue: #1334), in individuals. 

### 4.3. Transfection with Small Interfering RNA (siRNA) and Overexpression Construct

Transfection of siRNAs or the overexpression construct has been performed using lipofectamine and plus reagents (Invitrogen, Carlsbad, CA, USA), as previously reported [40]. Oligonucleotides specific for PTEN (sc-29459 and 5728-1) and PLK1 (sc-36277 and 5347-1) have been obtained from Santa Cruz Biotechnology (Santa Cruz, CA, USA) and Bioneer (Daejeon, Korea). Scrambled control (sc-37007) and PLK1 overexpression construct (sc-400411) have been purchased from Santa Cruz Biotechnology. The efficacy of siRNA or overexpression construct transfection has been confirmed by Western blot analysis of corresponding proteins (original figures can be found at Appendix A).

### 4.4. Phosphor-RTK Array

To demonstrate the intracellular signaling mechanism by si-PTEN transfection in Panc-1 cells, a phospho-RTK array kit (ARY001B, R&D Systems, Minneapolis, MN, USA) has been used according to the manufacturer’s instructions.

### 4.5. Measurement of Cell Viability

To evaluate cell viability with si-PTEN transfection or SF1670 treatment, a WST-1 solution (2-(4-Iodophenyl)-3-(4-nitrophenyl)-5-(2,4-disulfophenyl)-2H-tetrazolium solution, Nalgene, Rochester, NY, USA) has been used as described previously [41]. AsPC-1, Capan-2, SNU-213, CFPAC-1, Panc-1, and Miapaca cells (5 × 10^3^/well) were seeded in 96-well plates (Nunc, Roskilde, Denmark). The cells were maintained in a culture medium for 18 h, which was followed by treatment with various dosages of SF1670 under a normal culture condition. The cells were incubated at 37 °C for an additional 72 h. Ten microliters of WST-1 solution were added to each well, and after 10 min of incubation at room temperature, the absorbance was measured at 450 nm using a microplate reader (Bio-Rad, Richmond, CA, USA).

### 4.6. Trans-Well Migration Assay

Migration assays were performed using a 24-well Trans-well apparatus (Corning, Corning, NY, USA), as described previously [27]. The cells were transfected with scrambled si-RNA or si-PTEN for 72 h under normal culture conditions. Polycarbonate filters were pre-coated with fibronectin (10 mg/L, Sigma-Aldrich, St. Louis, MO, USA) in phosphate-buffered saline for 30 min at room temperature. The lower chamber was filled with 500 μl of RPMI-1640 medium containing 10% serum. After suspending the scrambled si-RNA or si-PTEN transfected cells (5 × 10^4^ cells/well) in serum-free RPMI-1640 medium, the cells were loaded into the upper chambers, which is followed by incubation in the previously mentioned medium for 6 h at 37 °C. The cells on the upper surface of the filter were removed using a cotton swab, and fixed and stained the filters using a 1% crystal violet solution. The eluted dye was measured at 560 nm in an ELISA reader (Bio-Rad, Richmond, CA, USA). 

### 4.7. Xenograft Tumour Model

BALB/c nude mice have been obtained from Orient (Seongnam, Korea) at 6 to 8 weeks of age. Panc-1 and AsPC-1 cells (1 × 10^7^) were injected subcutaneously into the right flank (Panc-1: *n* = 18, AsPC-1: *n* = 18), as previously described [42]. Once the tumours achieved a size of approximately 200 mm^3^, mice were randomized to three experimental groups of treatment: mock, 10 mg/kg, or 30 mg/kg of SF1670. Tumour volume (V) was calculated as 0.523 LW^2^ (L = length, W = width). Body weight was recorded regularly every two days. Animal care and experiments have been carried out in accordance with guidelines approved by the Animal Bioethics Committee of Jeju National University (2016-0049). 

### 4.8. GSE Data-Set Analysis 

Gene expression data (E-MEXP-2780, GSE21501, GSE57495, GSE71729, GSE84219, PACA-AU_[Pancreatic Cancer-AU], GSE17891, GSE47368, GSE62452, GSE79668, PAAD-US_Pancreatic Cancer-TCGA_US, and PAEN-AU_[Pancreatic Cancer Endocrine Neoplasms-AU]) with prognostic information have been downloaded from the Gene Expression Omnibus, ArrayExpress, and the ICGC data portal. Each probe was converted to EntrezID. Several probes for the same EntrezID were averaged. Quantile-quantile normalization was applied to all samples to remove batch effects. To determine the prognostic value of a gene, samples were divided into two groups using the median gene expression level as the threshold. The log-rank test was then performed using Graph Prism version 5, and several log-rank p-values from the data sets were integrated into a single p-value using Liptak’s weighted z-score method using the square root of sample numbers as weight, which was described previously [43]. Prognostic values associated with PLK1 and PTEN expression were determined. All patients have been divided into two groups based on PLK1 expression using the average PLK1 expression as the threshold. Patients have been further divided into a PTEN-low group and a PTEN-high group in each PLK1-low and PLK1-high group. Probabilities of patient deaths in PTEN-low and PTEN-high groups have been determined using the log-rank test.

### 4.9. Iterative Patient Partitioning (IPP) Method 

We have systematically explored the genetic determinants associated with PTEN-based prognosis. We have used the iterative patient partitioning method (IPP), which is a nonparametric version of the log-rank test, to avoid bias associated with the conventional log-rank test focused on a fixed patient partition. For each marker gene, patients have been stratified into two groups (low and high) according to the marker gene expression. IPP scores of PTEN in the low and high groups were calculated and compared. All genes were used as markers to stratify patients into two groups. A total of 11 pancreatic data-sets were used. The average of IPP scores of PTEN in data sets after excluding those with the largest and smallest IPP scores have been used to obtain the final summarized IPP score of PTEN in each group with low and high levels of markers. 

### 4.10. Statistical Analysis

All data have been presented as means ± standard deviation. Levels of significance for comparisons between two independent samples have been determined using tge Student’s *t*-test. Groups have been compared using one-way ANOVA with Tukey’s *post hoc* test applied to significant main effects (SPSS 12.0K for Windows; SPSS Inc., Chicago, IL, USA).

## 5. Conclusions

This is the first report to verify PTEN as the therapeutic target in human pancreatic cancer. Targeting PTEN shows differential dependency on malignant features of pancreatic cancer through the expression levels of PLK1. The expression level of PLK1 renders differential clinical outcomes of PTEN highly expressed pancreatic cancer patients. Soluble PLK1 is the blood biomarker in pancreatic cancer. PTEN and PLK1 is the companion biomarker in pancreatic cancer.

## Figures and Tables

**Figure 1 cancers-12-00277-f001:**
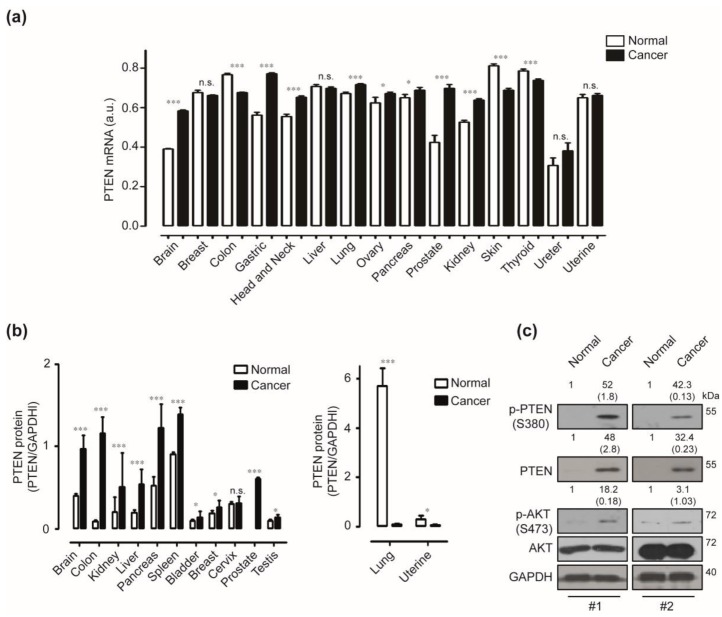
Expressions of phosphatase and tensin homolog (PTEN) in human pancreatic cancer. (**a**) Transcriptional levels of *PTEN* in various cancers and matched normal tissues in the *Oncopression* databases have been analyzed. (a.u. indicates arbitrary unit using the UPCs method, *p*-value evaluated with Student’s *t* test, * *p* < 0.05, *** *p* < 0.001, n.s. means non-significant). (**b**) Human Normal Tissue Blot I and Human Tumor Tissue Blot I have been used to determine the expression of PTEN in various normal and tumour tissues. GAPDH has been used as a control. Data represent two individual experiments. Relative pixel intensity for PTEN has been measured using densitometry analysis (PTEN/GAPDH) using ImageJ analysis software. (**c**) Protein expressions of p-PTEN, PTEN, p-AKT, and AKT in pancreatic cancers and normal pancreas has been analyzed using the Western blot. GAPDH has been used as a control (Normal indicates a normal pancreas sample. Cancer indicates a pancreatic cancer sample. #1 and #2 represent separate samples. Data are representative of three individual experiments).

**Figure 2 cancers-12-00277-f002:**
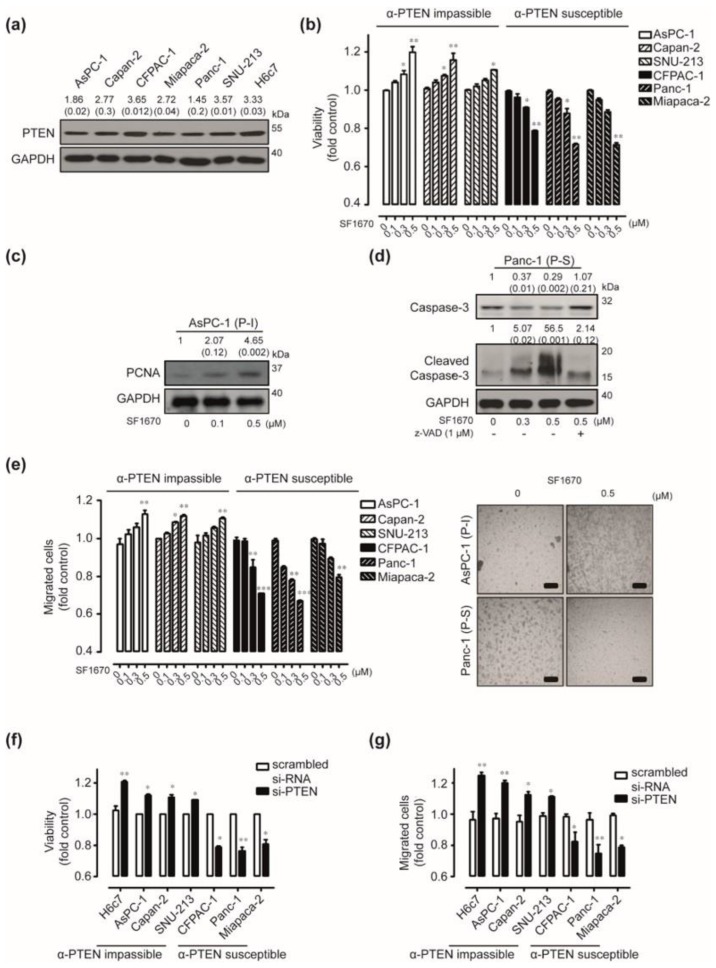
In vitro effects of phosphatase and tensin homolog (PTEN) blockade in human pancreatic cancer. (**a**) PTEN proteins in various human pancreatic cancer cells and H6c7 cells have been detected using Western blot. GAPDH was used as a control. Relative pixel intensities of PTEN were measured using ImageJ analysis software (PTEN/GAPDH). Data represent three individual experiments. (**b**) AsPC-1, Capan-2, Miapaca-2, SNU-213, CFPAC-1, Panc-1, and Miapaca-2 cells were incubated with varying doses of SF1670 for 72 h. The viability was measured by the WST-1 assay (*n* = 3, Tukey’s *post-hoc* test has been used to detect significant differences in ANOVA, *p* < 0.0001. Asterisks indicate a significant difference compared with 0% inhibition. Data represent three individual experiments, * *p* < 0.05, ** *p* < 0.01, *** *p* < 0.001). (**c**) AsPC-1 cells were incubated with varying doses of SF1670 for 72 h. Cell lysates were subjected to Western blot analysis using antibodies against PCNA and GAPDH. (**d**) Panc-1 cells were incubated either in the absence or presence of SF1670 (0, 0.3, and 0.5 µM) and/or z-VAD (1 µM) for 72 h. Cell lysates were subjected to Western blot analysis using antibodies against total caspase-3, cleaved-caspase-3, and GAPDH. Relative pixel intensity for PCNA and caspase-3 was measured by densitometry (PCNA/GAPDH or caspase-3/GAPDH) using ImageJ analysis software. (**e**) Left: AsPC-1, Capan-2, Miapaca-2, SNU-213, CFPAC-1, Panc-1, and Miapaca-2 cells were incubated with varying doses of SF1670 for 6 h. The migration activities have been evaluated using the Transwell-migration assay (*n* = 3, Tukey’s *post-hoc* test has been used to detect significant differences in ANOVA, *p* < 0.0001, asterisks indicate a significant difference compared with 0% inhibition. Data represent three individual experiments, * *p* < 0.05, ** *p* < 0.01, *** *p* < 0.001). Right: representative image of Trans-well migration assay of AsPC-1 and Panc-1 cells (scale bar = 50 µm). (**f**) H6c7, AsPC-1, Capan-2, Miapaca-2, SNU-213, CFPAC-1, Panc-1, and Miapaca-2 cells were transfected with scrambled or PTEN-specific siRNA for 72 h. The viability was measured by the WST-1 assay (*p*-value evaluated with Student’s *t* test, and data represent three individual experiments, ** p* < 0.05, *** p* < 0.01). (**g**) H6c7, AsPC-1, Capan-2, Miapaca-2, SNU-213, CFPAC-1, Panc-1, and Miapaca-2 cells were transfected with scrambled or PTEN-specific siRNA. After 48 h of transfection, the cells were exposed to serum-starved conditions. After 24 h of serum starvation, migrated cells were evaluated using the Transwell-migration assay for 6 h (*p*-value evaluated with Student’s *t* test, and data represent three individual experiments, * *p* < 0.05, ** *p* < 0.01).

**Figure 3 cancers-12-00277-f003:**
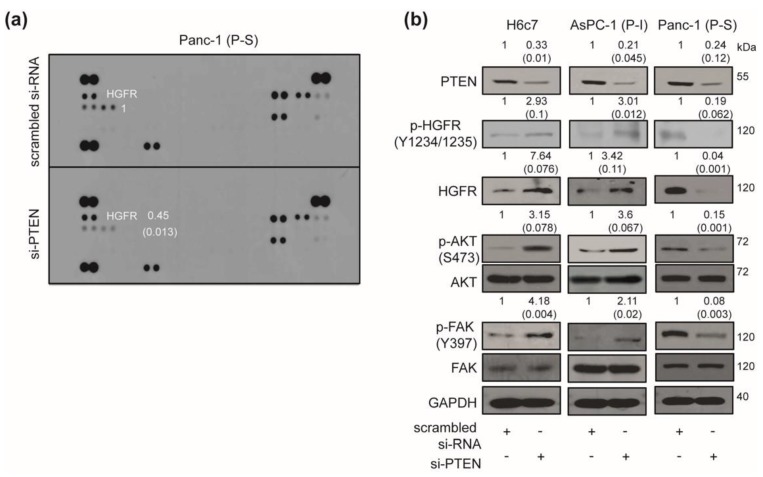
Effect of phosphatase and tensin homolog (PTEN) blockade on intracellular signaling pathway. (**a**) Panc-1 cells have been transfected with scrambled or PTEN-specific siRNA for 48 h. Human phospho-RTK array was used to determine differences in scrambled or PTEN-specific siRNA transfection. Relative pixel intensity for p-HGFR (p-HGFR/positive spot) was measured by densitometry analysis using ImageJ analysis software. Data represents two individual experiments. (**b**) H6c7, AsPC-1, and Panc-1 cells were transfected with scrambled or PTEN-specific siRNA for 48 h and the cell lysates were subjected to a Western blot using specific antibodies targeting PTEN, p-HGFR (Y1234/1235), HGFR, p-AKT (S473), AKT, p-FAK (Y397), FAK, and GAPDH. Relative pixel intensities have been measured by densitometry analysis using ImageJ analysis software. Data represent three individual experiments.

**Figure 4 cancers-12-00277-f004:**
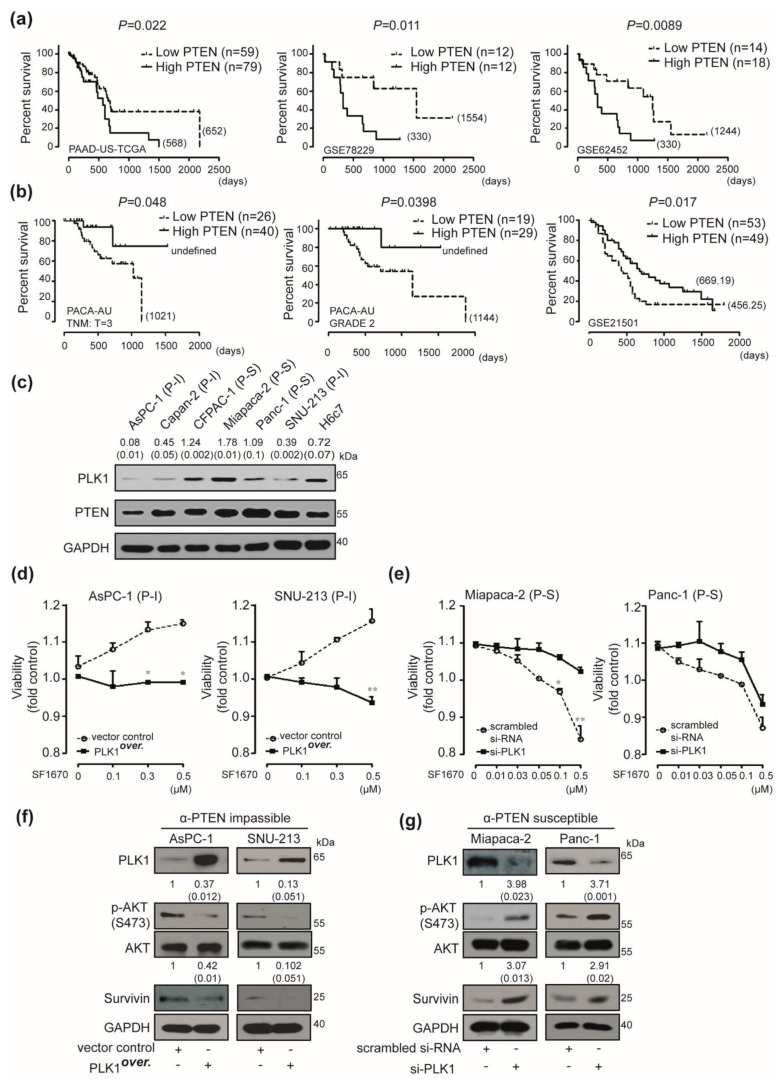
Expression of polo-like kinase 1 (PLK1) as a companion biomarker with phosphatase and tensin homolog (PTEN) expression in pancreatic cancer. (**a**) Overall survival was analyzed using Kaplan-Meier curves depending on the differential expression of PTEN in PTEN-adverse data-sets, PAAD-US-TCGA, GSE78229, and GSE62452 (*p* value was calculated using Log-rank (Mantel-Cox) Test). (**b**) Overall survival was analyzed using Kaplan-Meier curves depending on the differential expression of PTEN in PTEN favorable data-sets, PACA-AU and GSE21501 (*p* value was calculated using Log-rank (Mantel-Cox) Test). (**c**) PLK1 and PTEN proteins in AsPC-1, Capan-2, Miapaca-2, SNU-213, CFPAC-1, Panc-1, Miapaca-2, and H6c7 cells were detected using the Western blot. GAPDH was used as a control. Relative pixel intensities of PLK1 were measured using ImageJ analysis software (PLK1/GAPDH). Data represent three individual experiments. (**d**) AsPC-1 and SNU-213 cells were transfected with a vector control or the PLK1 overexpression construct. After 48 h of transfection, AsPC-1 and SNU-213 cells were incubated with different doses of SF1670 for an additional 72 h. Viability was measured by the WST-1 assay (*n* = 3, Tukey’s *post hoc* test has been used to determine the significant group effects in ANOVA, *p* < 0.0001, asterisks indicate a significant difference compared with 0% inhibition, * *p* < 0.05, ** *p* < 0.01). (**e**) Miapaca-2 and Panc-1 cells were transfected with scrambled or PLK1-specific siRNA. At 48 h post-transfection, cells were incubated with different doses of SF1670 for an additional 72 h. Viability was measured by the WST-1 assay (*n* = 3, Tukey’s *post hoc* test has been used to determine the significant group effects in ANOVA, *p* < 0.0001. Asterisks indicate a significant difference compared with 0% inhibition, * *p* < 0.05, ** *p* < 0.01). (**f**) After 48 h of transfection with the vector control or PLK1 overexpression construct, AsPC-1 and SNU-213 cell lysates were subjected to Western blot analysis using antibodies specific for PLK1, PTEN, p-AKT, AKT, and Survivin. GAPDH was used as a control. (**g**) Miapaca-2 and Panc-1 cells were transfected with scrambled or PLK1-specific siRNA. After 48 h of transfection, Miapaca-2 and Panc-1 cell lysates were subjected to immunoblot analysis using antibodies specific for PLK1, PTEN, p-AKT, AKT, and Survivin. GAPDH was used as a control.

**Figure 5 cancers-12-00277-f005:**
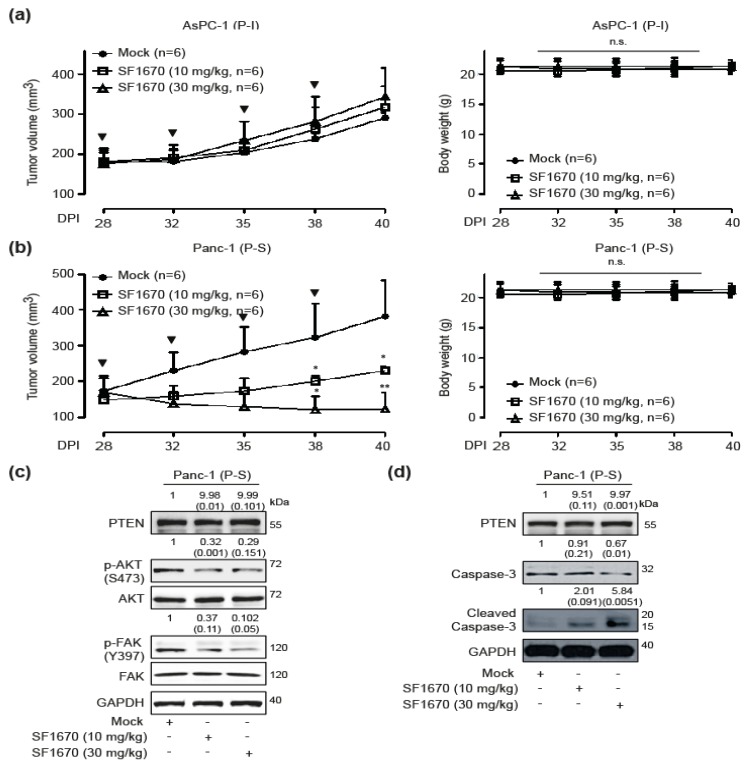
In vivo effects of phosphatase and tensin homolog (PTEN) blockade. (**a**) Left, the effect of different dosages of SF1670 (10 mg/kg and 30 mg/kg) in Aspc-1 xenograft models (control group: *n* = 6, 10 mg/kg: *n* = 6, and 30 mg/kg: *n* = 6) were measured for 40 days using the formula: V = 0.523 LW^2^ (L = length, W = width). Bold arrows indicate the time of SF1670 injection (Tukey’s post-hoc test has been used to determine significant group effects in ANOVA, *p* < 0.0001. Asterisks indicate a significant difference between the control group and the SF1670-injected group). Right, the body weight in each group was measured regularly. (**b**) Left, the effect of different dosages of SF1670 (10 mg/kg and 30 mg/kg) in Panc-1 xenograft models (control group: *n* = 6, 10 mg/kg: *n* = 6 and 30 mg/kg: *n* = 6) were measured for 40 days using the formula: V = 0.523 LW^2^ (L = length, W = width). Bold arrows indicate the time of SF1670 injection (Tukey’s post-hoc test has been used to determine significant group effects in ANOVA, *p* < 0.0001. Asterisks indicate a significant difference between the control group and the SF1670-injected group, * *p* < 0.05, ** *p* < 0.01). Right, the body weight in each group was measured regularly. (**c**) The lysates derived from Panc-1 xenograft tumour samples were tested for p-AKT, AKT, p-FAK (Y397), FAK, and GAPDH by Western blot analysis. Relative pixel intensities have been measured by densitometry analysis using ImageJ analysis software. Data is representative of three individual experiments. (**d**) The lysates derived from Panc-1 xenograft tumour samples were tested caspase-3, cleaved caspase-3, and GAPDH by Western blot analysis. Relative pixel intensities have been measured by densitometry analysis using ImageJ analysis software. Data is representative of three individual experiments.

**Figure 6 cancers-12-00277-f006:**
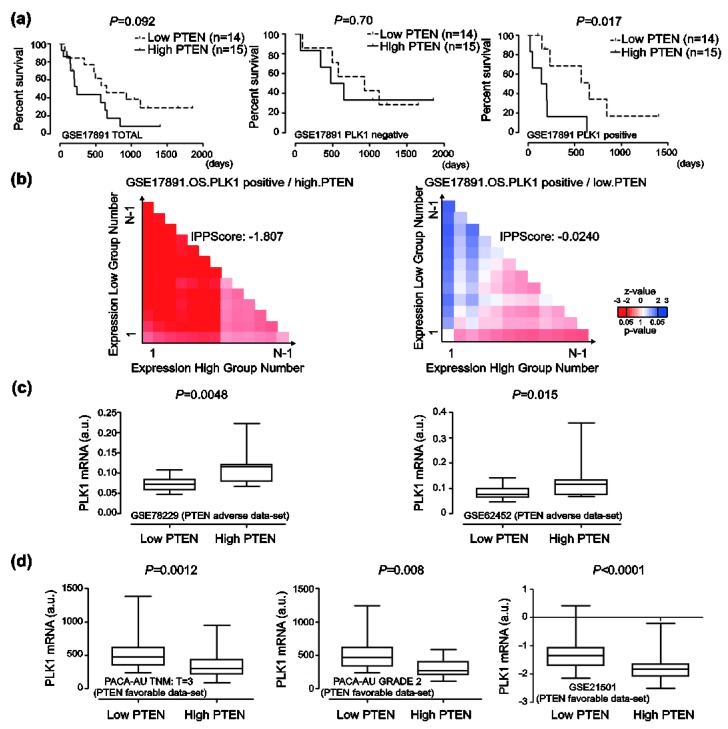
Prognosis associated with phosphatase and tensin homolog (PTEN) and polo-like kinase 1 (PLK1) expressions in human pancreatic cancer. (**a**) Overall survival of pancreatic cancer patients was analyzed using Kaplan-Meier curves depending on the PLK1 and PTEN expressions in GSE17891 data-set (*p* value was calculated using the Log-rank (Mantel-Cox) Test). (**b**) Prognostic relevance was analyzed using the log-rank test and its non-parametric version of curves on the differential expression of PLK1 and PTEN in the GSE17891 data-set. (**c**) Transcriptional levels of *PLK1* in low PTEN and high PTEN groups were analyzed using GSE78229 and GSE62452 data sets. (a.u. indicates arbitrary unit using the UPCs method. The *p*-value was evaluated with a Student’s *t* test). (**d**) Transcriptional levels of *PLK1* in low PTEN and high PTEN groups were analyzed using PACA-AU and GSE21501 data sets. (a.u. indicates an arbitrary unit using the UPCs method. The *p*-value was evaluated with a Student’s *t* test).

**Figure 7 cancers-12-00277-f007:**
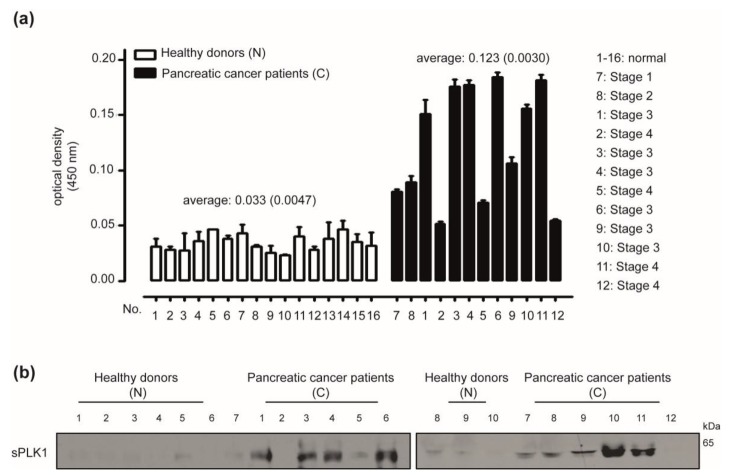
Selection of Polo-like kinase1 (PLK1) using easily applicable techniques and whole blood samples of pancreatic cancer patients. (**a**) soluble PLK1s (sPLK1) were selected by a sandwich ELISA analysis using the same volumes of whole bloods samples from 16 healthy donors (N1–N16) and 12 pancreatic cancer patients (C1–C12). (**b**) Soluble PLK1s (sPLK1) were selected by Western blot analysis using the same volumes of whole blood samples from 10 healthy donors (N1–N10) and 12 pancreatic cancer patients (C1–C12).

**Table 1 cancers-12-00277-t001:** IPPScores compared with PLK1 expression levels in human pancreatic cancer.

Data-Set	SQRT Low Group (*N*)	SQRT High Group (*N*)	PLK1 Low Group IPPScore	PLK1 High Group IPPScore
E-MEXP-2780	4.3588 (19)	3.3166 (11)	−1.02099	−1.1888
GSE17891	3.6055 (13)	3.7416 (14)	−0.024019	−1.8071
GSE62452	7 (49)	4.1231 (17)	0.502234	0.594234
GSE21501	7.6157 (58)	6.6332 (44)	0.533184	−0.247134
GSE57495	5.6568 (32)	5.5677 (31)	−0.790192	−0.710717
GSE71729	8.3066 (69)	7.4833 (56)	−0.899932	−0.188088
GSE84219	4.1231 (17)	3.6055 (13)	1.29252	0.809448
PAAD-US	9.2195 (85)	7.2801 (53)	−1.09615	−1.23564
PACA-AU	6.9282 (48)	5.5677 (31)	0.85567	−1.15497
Liptak’s *z*-value			−0.392674332	−1.68977768
Liptak’s *p*-value			0.347280015	0.045535248

E-MTAB-6134 and GSE79668 were excluded from the final IPP score because E-MTAB-6134 expressed the maximum IPPScore (2.09) and GSE79668 exhibited the minimum IPPScore (−2.15).

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
