# Peer review of "Differential Dependency of Human Pancreatic Cancer Cells on Targeting PTEN via PLK 1 Expression"

_cancers, 2020, doi:10.3390/cancers12020277_

Round 1

Reviewer 1 Report

Thanks for providing answers to part of the questions. 

Results shown in Fig 1c (PTEN phosphorylation and phosphorylation of AKT) suggest to me that PTEN could be catalytically dead.

This manuscript is a resubmission of an earlier submission. The following is a list of the peer review reports and author responses from that submission.

Round 1

Reviewer 1 Report

PTEN is well known for having tumor suppressor activity, and PTEN loss has been found to be associated with cancer recurrence and increased metastatic burden for several cancers.

In this article, the authors propose to use PTEN blockade therapy using PLK1 expression as a companion biomarker. Here authors report one group of human pancreatic cancer cell lines to be PTEN blockade susceptible while another group being PTEN blockade impassible. Analyzing different GEO data sets, authors reported PTEN expression to be positively correlated with disease prognosis in certain data sets while in other data sets PTEN expression is negatively correlated with disease progression. Additionally, authors report PLK1 expression to be uniquely associated with the differential response of pancreatic cancer cells to PTEN blockage and suggest the possibility of selecting patient population for PTEN blockage therapy based on PLK1 expression which can be detected in the blood. Is there any correlation between Kras mutational status and/or PTEN mutation among these different patient data-set groups with different PTEN prognostic values?

Comment 1: Authors should specify pancreatic ductal adenocarcinoma rather than general pancreatic cancer which also includes endocrine pancreatic tumors, acinar cell carcinoma, etc.

Comment 2: Title should include "human pancreatic cancer cell lines" instead of "human pancreatic cancers"

Comment 3:Throughout the paper, the authors identified human cancer cell lines as human cancer or human cancer cells. Human cancer or human cancer cells would mean patient-derived primary cell culture.

Comment 4: Lines 308 - 313 " Treatments with anti-pancreatic cancer agents........."

Chemotherapy drugs or chemotherapeutic can be used instead of "anti-pancreatic cancer agents". Also, these sentences overstate PTEN blockade not having toxicity issues. H6C7 is an immortalized cell line derived from normal human pancreatic duct epithelial. PTEN blockade toxicity on pancreatic acinar cells can neither be judged from the experiments performed in this article nor be compared with the toxicity from currently used chemo-therapeutics. 

Comment 5: Most of the figures are of low resolution and. The authors need to improve the quality of the figures.

Minor comments- 

Comment 6: Lines 86-89: Viability data has been presented as "fold change" for some samples and "% change" for others. Sticking to one format will be helpful.

Comment 7: In my opinion, some sections of results and discussion sections are difficult to comprehend due to sentence structures and word choices. The authors should consider restructuring several sentences throughout the paper for clarity. Few examples -  

Lines 89 - 91 - "To determine the altered.........treatment".

Lines 294 - 295 "Although increasing numbers of PTEN......"

Comment 8: Spell check needed. For example line 184 " Enhanced PLK1 expression.......and survivin in ...... cells"

Author Response

Reviewer 1

Comments and Suggestions for Authors

PTEN is well known for having tumor suppressor activity, and PTEN loss has been found to be associated with cancer recurrence and increased metastatic burden for several cancers.

In this article, the authors propose to use PTEN blockade therapy using PLK1 expression as a companion biomarker. Here authors report one group of human pancreatic cancer cell lines to be PTEN blockade susceptible while another group being PTEN blockade impassible. Analyzing different GEO data sets, authors reported PTEN expression to be positively correlated with disease prognosis in certain data sets while in other data sets PTEN expression is negatively correlated with disease progression. Additionally, authors report PLK1 expression to be uniquely associated with the differential response of pancreatic cancer cells to PTEN blockage and suggest the possibility of selecting patient population for PTEN blockage therapy based on PLK1 expression which can be detected in the blood.

Is there any correlation between Kras mutational status and/or PTEN mutation among these different patient data-set group s with different PTEN prognostic values?

Response:

As commented, we tried to validate the correlation between Kras mutational status and/or PTEN mutation among these different patient data-set groups with different PTEN prognostic values. However, our data-sets had no prognostic information to Kras and/or PTEN mutational status. As per commented, the correlation between Kras mutational status and/or PTEN mutation with different PTEN prognostic values in pancreatic cancer patients would be critical for pancreatic cancer biology, therefore, we decided to figure out the critical correlation between Kras and/or PTEN mutational status and different PTEN prognostic values updating our data-sets.

Comment 1: Authors should specify pancreatic ductal adenocarcinoma rather than general pancreatic cancer which also includes endocrine pancreatic tumors, acinar cell carcinoma, etc.

 Response:

As commented, we specified pancreatic ductal adenocarcinoma rather than general pancreatic cancer in Line 28-29 of page 1.

Comment 2: Title should include "human pancreatic cancer cell lines" instead of "human pancreatic cancers"

Response:

In the present study, we demonstrate the effect of PTEN blockade on human pancreatic cancer based on PLK1 expression adducing in silico, in vitro, in vivo, and clinical evidences. In silico and clinical data include the information of pancreatic cancer from pancreatic cancer patients, therefore, we decided to transform the “cells” rather than “cell lines” in the title of this manuscript in Line 2-3 of page 1.

Comment 3: Throughout the paper, the authors identified human cancer cell lines as human cancer or human cancer cells. Human cancer or human cancer cells would mean patient-derived primary cell culture.

 Response:

In the present study, we demonstrate the effect of PTEN blockade on human pancreatic cancer based on PLK1 expression adducing in silico, in vitro, in vivo, and clinical evidences. In silico and clinical data include the information of pancreatic cancer from pancreatic cancer patients, therefore, we transform the title of this manuscript as Different Dependency of Human Pancreatic Cancer cells on Targeting PTEN via PLK1 Expression.

Comment 4: Lines 308 - 313 " Treatments with anti-pancreatic cancer agents........."

Chemotherapy drugs or chemotherapeutic can be used instead of "anti-pancreatic cancer agents".

 Response:

As commented, we corrected “anti-pancreatic cancer agents” to “chemotherapeutics” in Line 310 of page 12.

Also, these sentences overstate PTEN blockade not having toxicity issues. H6C7 is an immortalized cell line derived from normal human pancreatic duct epithelial. PTEN blockade toxicity on pancreatic acinar cells can neither be judged from the experiments performed in this article nor be compared with the toxicity from currently used chemo-therapeutics. 

 Response:

Firstly, we highlighted the biological meaning of PTEN in pancreatic ductal adenocarcinoma cells as your comment. In Figure 2a and Supplementary Figure 2a-c, H6c7, human normal pancreatic duct epithelial cell had the relatively high cellular PTEN expression level and silencing PTEN increased the viability and migration compared to those of control. In view of anti-pancreatic cancer therapy, the results following PTEN blockade in H6c7 would be potential features of normal pancreas cells and/or pancreas surrounding normal cells. Based on these meanings, we proposed the toxicity issues using human pancreatic cancer cells and H6c7 human human normal pancreatic duct epithelial cell. We described these contents in Line 312-315 of page 12 newly containing “potentially”.

Comment 5: Most of the figures are of low resolution and. The authors need to improve the quality of the figures.

 Response:

As commented, we prepared the more qualified high resolution figures.

Minor comments- 

Comment 6: Lines 86-89: Viability data has been presented as "fold change" for some samples and "% change" for others. Sticking to one format will be helpful.

 Response:

As commented, we unified to one format as “fold control”.

Comment 7: In my opinion, some sections of results and discussion sections are difficult to comprehend due to sentence structures and word choices. The authors should consider restructuring several sentences throughout the paper for clarity.

 Response:

As commented, English in this document has been checked by at least two professional native speakers of English.

Few examples -  

Lines 89 - 91 - "To determine the altered.........treatment".

Response:

As commented, we corrected the sentence to “To determine the effect of targeting PTEN on viability, we have investigated the levels of proliferating cell nuclear antigen (PCNA) in AsPC-1 cells, which were most increased on viability by SF1670 treatment” in Line 89-91 of page 3.

Lines 294 - 295 "Although increasing numbers of PTEN......"

 Response:

As commented, we corrected the sentence to “Despite increasing the features of PTEN as non-tumour suppressor in various cancers” in Line 296-297 of page 12.

Comment 8: Spell check needed. For example line 184 " Enhanced PLK1 expression.......and survivin in ...... cells"

 Response:

As commented, we corrected the sentence to “Enhanced PLK1 expression decreased the levels of AKT phosphorylation and survivin expressions” in Line 185-186 of page 6.

Submission Date

07 November 2019

Date of this review

15 Nov 2019 10:43:26

Reviewer 2 Report

Lee and group have studied the role of PTEN in pancreatic cancer. Authors have used databases and experimental techniques to establish PTEN/PLK1 as a prognostic marker. The study is good and can be accepted for publication after revision. My comments are as follows.

Please rewrite the abstract for clarity. Figure 1b: apply statistics Figure 2e is not clear. Use better images for migration assay. Please write a line in the result section where how authors classified the pancreatic cancer cell lines on PTEN impassible and susceptible groups? Use lower gapdh exposures for figure 3b lower panel. Figure 7: Include a graph showing the average expression of a healthy donor and PDAC patients in ELISA assay. Can you also check if PTEN and PLK1 are correlated in the TCGA database? What is the spearman constant, p-value etc? What happens to PTEN levels after overexpression or knockdown of PLK1 and vice versa. don’t use grayscale in figures it becomes difficult to see.

Author Response

Reviewer 2

Comments and Suggestions for Authors

Lee and group have studied the role of PTEN in pancreatic cancer. Authors have used databases and experimental techniques to establish PTEN/PLK1 as a prognostic marker. The study is good and can be accepted for publication after revision. My comments are as follows.

Please rewrite the abstract for clarity.

Response:

As commented, we re-wrote the abstract to clarify our manuscript in Line 14-15 and Line 16-18 of page1.

Figure 1b: apply statistics

Response:

As commented, we prepared statistics in Figure 1b.

Figure 2e is not clear. Use better images for migration assay.

Response:

As commented, we prepared more qualified images by changing resolution in Figure 2e.

Please write a line in the result section where how authors classified the pancreatic cancer cell lines on PTEN impassible and susceptible groups?

Response:

As commented, we added the rationale of pancreatic cancer cells classification by targeting PTEN in Line 104-106 of page 3.

Use lower gapdh exposures for figure 3b lower panel.

Response:

As commented, we prepared lower GAPDH exposures of Figure 3b.

Figure 7: Include a graph showing the average expression of a healthy donor and PDAC patients in ELISA assay.

Response:

As commented, we prepared the average values of healthy donors and pancreatic cancer patients in Figure 7a.

Can you also check if PTEN and PLK1 are correlated in the TCGA database?

Response:

As commented, we evaluated the expressed correlation between PTEN and PLK1, however, there was no significant correlation between PTEN and PLK1 expression. According to our results and preliminary evaluations, we more strongly considered the PTEN and PLK1 to companion biomarker better than biological expression counter partner in natural pancreatic cancer environments.

What is the spearman constant, p-value etc?

Response:

In the present study, we performed statistical analysis using Student’s t-test for two independent samples and one-way ANOVA with Tukey’s post hoc test for groups. In case of survival curve analysis, we used the Log-rank (Mantel-Cox) Test.

What happens to PTEN levels after overexpression or knockdown of PLK1 and vice versa.

Response:

As commented, we performed Western blot analysis of PTEN expression following overexpression or knockdown of PLK1 in 6 pancreatic cancer cells. On the contrary, we also performed Western blot analysis of PLK1 expression following overexpression or knockdown of PTEN in 6 pancreatic cancer cells. According to our Western blot analysis, there was no significant correlation between PTEN and PLK1 expression similar to pattern of mRNA expression levels. Therefore, we more clearly considered the correlation of PTEN and PLK1 expression to companion biomarker better than biological expression counter partner in natural pancreatic cancer environments.

don’t use grayscale in figures it becomes difficult to see.

Response:

In preparing Figure 2e, we have saved the all images using grayscale. To help see intuitively in Figure 2e we upgraded the resolution of Figure 2e.

Reviewer 3 Report

The aim of the present study is to explore the function of PTEN expression in human pancreatic cancer and to use PTEN blockade based on PLK1 expression as a cancer therapeutic strategy. 

Fig 1c: what is the phosphorylation of AKT on these samples? Is this PTEN active/phosporylated?

The authors, to determine the functional role of PTEN expression in human pancreatic cancers, used western blot to analyse the endogenous expression of PTEN in six pancreatic cancer cell lines and in the normal pancreatic duct epithelial H6c7 cells.

 line 84: "PTEN was highly expressed in both human pancreatic cancer cells and H6C7 cells (Figure 2a). Highly expressed compared to what? what is the range low-high used to determine that PTEN is highly expressed?

SF1670 inhibitor as been used to downregulate PTEN activity, however  it has to be kept in mind that this is not a specific inhibitor but it has effects also on other enzymes. 

fig 3: Downregulation of PTEN expression leads to upregulation of AKT phosphorylation in H6c7 and AsPC-1 cells whereas there is no effect on AKT phosphorylation in Panc-1 cells. How do you explain that? what would be the mechanism?

Moreover, reduced phosphorylation of HGFR is described as consequences of PTEN knockdown. However it is clear from fig 3b that there is an effect on the abundance of HGFR protein itself which would explain the reduced level of phosphorylation detected. 

Fig4 f and g: what is the level of PTEN after si-PLK1?

Fig5 c and d: please provide western blot for PTEN  

In all figures with western blots, the phospho antibodies used should be clearly labelled with the phospho site. P-AKT is never labelled. From the M&M it looks like p-AKT S473 was used, however as T308 correlates better with PIP3 levels, it would be good to look at it as well. 

Supplementary material

Figure S1

Is the molecular weight of PTEN different in different normal tissues? 

In all westernblot what are the numbers on each band?

Author Response

Reviewer’ 3 comments

The aim of the present study is to explore the function of PTEN expression in human pancreatic cancer and to use PTEN blockade based on PLK1 expression as a cancer therapeutic strategy.

Fig 1c: what is the phosphorylation of AKT on these samples? Is this PTEN active/phosporylated?

Response:

In Figure 1c, upper bands indicated the expression levels of PTEN, not AKT, of commercially buying normal pancreas and pancreatic cancer lysates isolated from healthy human and pancreatic cancer patients, respectively. The antibody of PTEN in this study (Cell signaling, #9188) was against for total PTEN, not phosphorylated.

The authors, to determine the functional role of PTEN expression in human pancreatic cancers, used western blot to analyse the endogenous expression of PTEN in six pancreatic cancer cell lines and in the normal pancreatic duct epithelial H6c7 cells.

line 84: "PTEN was highly expressed in both human pancreatic cancer cells and H6C7 cells (Figure 2a). Highly expressed compared to what? what is the range low-high used to determine that PTEN is highly expressed?

Response:

As commented, the representation as “highly expression” potentially would be confused to readers, therefore we corrected the “high” to “abundant” in this manuscript. In Figure 2 a-g, we tried to show the similarly abundant expression of PTEN in 6 pancreatic cancer cells and H6c7 cells and different responses to targeting PTEN in in 6 pancreatic cancer cells and H6c7 cells.

SF1670 inhibitor as been used to downregulate PTEN activity, however it has to be kept in mind that this is not a specific inhibitor but it has effects also on other enzymes.

 Response:

As commented, SF1670 is not the specific inhibitor against PTEN activity. Therefore, we additionally performed the si-PTEN transfection in pancreatic cancer cells and H6c7 cells to complement the deficient of SF1670 usage. SF1670 treatment and silencing PTEN using si-RNA showed similar patterns on viability and migration of pancreatic cancer cells and H6c7 cells.

fig 3: Downregulation of PTEN expression leads to upregulation of AKT phosphorylation in H6c7 and AsPC-1 cells whereas there is no effect on AKT phosphorylation in Panc-1 cells.

How do you explain that? what would be the mechanism?

 Response:

As commented, establishment of fundamental signaling pathway of different feature by same targeting in pancreatic cancer cells was the one of the critical factors of this manuscript. We showed the promising therapeutic target as PTEN blockade in case of PTEN blockade susceptible group according to the present study, however, we still try to investigate the signaling pathway using in silico, in vitro, and clinical evidences to establish the novel mechanism in pancreatic cancer cells by future works. We authors considered it to the limitation of this manuscript and described these contents in Line 328-330 of page 13.

Moreover, reduced phosphorylation of HGFR is described as consequences of PTEN knockdown. However it is clear from fig 3b that there is an effect on the abundance of HGFR protein itself which would explain the reduced level of phosphorylation detected.

Response:

In Figure 3b, silencing PTEN induced the alteration of HGFR expression, not phosphorylated HGFR. We considered the potentially initial turning feature was to alteration of HGFR expression using RTK array. PTEN blockade impassible or susceptible group showed notably different features following si-PTEN transfection. However, future work should be performed to establish the fundamental signaling mechanism by PTEN targeting.

Fig4 f and g: what is the level of PTEN after si-PLK1?

Response:

As commented, we performed Western blot analysis of PTEN expression following overexpression or knockdown of PLK1 in 6 pancreatic cancer cells. On the contrary, we also performed Western blot analysis of PLK1 expression following overexpression or knockdown of PTEN in 6 pancreatic cancer cells. According to our Western blot analysis, there was no significant correlation between PTEN and PLK1 expression similar to pattern of mRNA expression levels. Therefore, we more clearly considered the correlation of PTEN and PLK1 expression to companion biomarker better than biological expression counter partner in natural pancreatic cancer environments.

Fig5 c and d: please provide western blot for PTEN

Response:

As commented, we provided Westen blot analysis for PTEN in Figure 5c-d. Administration of SF1670 did not affect the expression levels of PTEN. We authors considered that targeting PTEN involved feature was induced by loss of function or gain of function inhibiting activity of PTEN using si-RNA transfection of SF1670 treatment.

In all figures with western blots, the phospho antibodies used should be clearly labelled with the phospho site. P-AKT is never labelled. From the M&M it looks like p-AKT S473 was used, however as T308 correlates better with PIP3 levels, it would be good to look at it as well.

Response:

As commented, we labelled the phosphor site of used antibodies.

Supplementary material

Figure S1

Is the molecular weight of PTEN different in different normal tissues?

Response:

Two independent membranes have different membrane size and interval of molecular ladder, so it could be look different. Comparing size marker and PTEN, PTEN protein size was simiar in normal and tumour tissues, respectively.

In all westernblot what are the numbers on each band?

Response:

Data are representative of two or three individual experiments. We described these information in Figure legends.

Round 2

Reviewer 1 Report

The authors responded to all the scientific concerns. Image quality can still be improved, specifically labels.  

Reviewer 2 Report

All my comments are answered and the manuscript can be accepted for publication. 

Reviewer 3 Report

The aim of the present study is to explore the function of PTEN expression in human pancreatic cancer and to use PTEN blockade based on PLK1 expression as a cancer therapeutic strategy.

-Fig 1c: what is the phosphorylation of AKT on these samples? Is this PTEN active/phosporylated?

The authors explained the figure, which was already clearly explained, but they did not answer the questions above: what is the phosphorylation of AKT on these samples? Is this PTEN active/phosporylated? Meaning, have you looked at AKT phosphorylation in these samples? And at PTEN phosphorylation? They express PTEN but its catalytic activity could be dead/or reduced.

-Reduced phosphorylation of HGFR is described as consequences of PTEN knockdown. However it is clear from fig 3b that there is an effect on the abundance of HGFR protein itself which would explain the reduced level of phosphorylation detected.

Comment referred to Panc-1: It is clear from fig 3b that there is an effect on the abundance of HGFR protein itself in Panc-1 which would explain the reduced level of phosphorylation detected.

-Fig4 f and g: what is the level of PTEN after si-PLK1?

How did you perform this analysis? Please provide western blot for PTEN after overexpression/knockdown of PLK1

-Fig5 c and d: please provide western blot for PTEN

The authors provided the requested western blot for PTEN, however the figures are no clearly labeled.

Please explain clearly - We authors considered that targeting PTEN involved feature was induced by loss of function or gain of function inhibiting activity of PTEN using si-RNA transfection of SF1670 treatment.

-In all figures with western blots, the phospho antibodies used should be clearly labelled with the phospho site. P-AKT is never labelled. 

The figures have no p-ATK site labels as asserted.